# Development of a Microfluidic Device to Form a Long Chemical Gradient in a Tissue from Both Ends with an Analysis of Its Appearance and Content

**DOI:** 10.3390/mi12121482

**Published:** 2021-11-29

**Authors:** Yasunori Tokuoka, Keiichi Kondo, Noboru Nakaigawa, Tadashi Ishida

**Affiliations:** 1Department of Mechanical Engineering, School of Engineering, Institute of Technology, Tokyo 226-8503, Japan; 2Department of Urology, Graduate School of Medicine, Yokohama City University, Tokyo 236-0004, Japan; kkurouro@yokohama-cu.ac.jp (K.K.); nakaigan@med.yokohama-cu.ac.jp (N.N.)

**Keywords:** microfluidics, tissue, H-shaped channel, long-term perfusion, internal analysis, appearance analysis, concentration gradient

## Abstract

Tissue assays have improved our understanding of cancers in terms of the three-dimensional structures and cellular diversity of the tissue, although they are not yet well-developed. Perfusion culture and active chemical gradient formation in centimeter order are difficult in tissue assays, but they are important for simulating the metabolic functions of tissues. Using microfluidic technology, we developed an H-shaped channel device that could form a long concentration gradient of molecules in a tissue that we could then analyze based on its appearance and content. For demonstration, a cylindrical pork tissue specimen was punched and equipped in the H-shaped channel device, and both ends of the tissue were exposed to flowing distilled and blue-dyed water for 100 h. After perfusion, the tissue was removed from the H-shaped channel device and sectioned. The gradient of the blue intensity along the longitudinal direction of the tissue was measured based on its appearance and content. We confirmed that the measured gradients from the appearance and content were comparable.

## 1. Introduction

Approximately 10 million people worldwide died of cancer in 2020 [1]. To study cancer, in vitro cancer cells are widely used in two-dimensional (2D) cultures [2,3]. Recently, cancer spheroids, which have three-dimensional (3D) structures, have been used. Thanks to their 3D structures, cancer spheroids have similar properties to cancer tissues in vivo; they have a high drug resistance and malignant transformation [4,5,6,7,8]. Although cancer spheroids have improved our understanding of cancer, they do not simulate in vivo cancer tissues in terms of intercellular communication.

In vivo cancer tissue is composed of not only cancer cells, but also normal cells in organs, stroma, blood vessels, etc. [9,10,11]. Interactions between diverse cells and stroma in tumors has been found to regulate cancer growth and metastasis [12,13,14,15,16,17,18]. Thus, insights into their interactions have been obtained through the use of tissue assays. Tissues to be assayed are cultured in vivo and their concentration gradient is caused by diffusion and consumption in a uniform culture medium. Perfusion-cultured tissues strongly affect metabolism [19,20]. In addition, hypoxic and normoxic cells, which are adjacent under oxygen and nutrient gradients in the centimeter scale, interact with each other to avoid apoptosis for several days, resulting in drug resistance in renal cancer [21,22,23]. However, tissue assays do not mature because tissues with biological distributions are ground for analysis after tissue experiments, resulting in only averaged data. Therefore, new tissue assays with both perfusion culture and concentration gradients are needed. To improve tissue assays and further our biological understanding, tissue culture techniques to control the culture medium around cancer tissues should be developed.

Microfluidic technology is capable of controlling the culture medium around cancer tissues. Microfluidic technology has achieved the highly precise control of the flow and concentration of a medium using its shapes and functional devices [24,25,26]. With such medium controls, microfluidic technology achieved the perfusion culture of tissues for 24 h and maintained the same metabolic functions as in vivo [27]. Additionally, high-throughput microfluidic devices have been used to culture some cubic and disk-shaped dissected tissues and examined multiple concentrations of drugs [28,29]. However, the tissues used in microfluidic devices are cultured in a culture medium with a uniform concentration distribution of oxygen and nutrients, while the gradient of oxygen and nutrients exists in the tissue in vivo. This is because the cancer tissue receives oxygen and nutrients from the capillaries, and a concentration gradient is formed according to the distance from them. When tissue contains blood vessels, longer diffusion can occur in it [30]. In addition, the analysis of tissue content is difficult, although interactions occur inside. This is because the tissue equipped in the microfluidic device cannot be removed after the experiment without becoming severely damaged. Therefore, microfluidic devices with both the formation of a concentration gradient in a tissue and the analysis of tissue content are required.

Here, we propose an H-shaped channel device with a mechanical seal to perfuse the medium for 100 h, in order to facilitate the formation of a concentration gradient in a tissue without any supply from the sidewall and to analyze the tissue inside. The H-shaped channel device has parallel channels with a connection channel between them. Tissue was installed in the connection channel. Deionized (DI) water and blue-dyed water were perfused into the parallel channels for 100 h, resulting in the formation of a long concentration gradient in centimeter order in the tissue. Temporal changes in the tissue color were observed from their appearance during perfusion, and the concentration gradient inside the tissue was measured from the slices after perfusion.

## 2. Materials and Methods

### 2.1. H-Shaped Channel Device and Its Peripherals

A schematic illustration of the H-shaped channel device is shown in Figure 1. The H-shaped channel device had two parallel channels and a long connection channel (Figure 1). The parallel channels were for the perfusion of two kinds of water, and the connection channel acted as a bridge between the parallel channels. A piece of tissue was placed in the connection channel and was well-sealed for the suppression of the water inflow from the parallel channels. By perfusing two kinds of water in the parallel channels, both ends of the tissue were exposed to different kinds of water, which resulted in the formation of a long gradient in the tissue without any disturbance from the influx of water from the sidewall. The H-shaped channel device, tissue, and holders were assembled as shown in Figure 2. To prevent water leakage, they were mechanically pressed by holders. Due to the mechanical seals, we were able to remove the tissue for the analysis of its contents after the experiments.

### 2.2. Design

The design of the H-shaped channel device is shown in Figure 3. The dimensions of the channels were designed based on the dimensions of the tissue (Figure 3a). The tissue used was 0.5 mm in diameter and 3 mm in length, which was decided based on clinical points of view. In this study, we used meat from a pig’s heart as a simulation of cancer tissue. The connection channel had the same dimensions as the tissue (0.5 mm in diameter × 3 mm in length) to close the connection channel with the tissue. The length of the tissue was adequate, because material diffusion in tissues has sometimes been reported to be of centimeter order [30]. The diameter was decided by the leakage test using simulated tissues (Appendix A). The width of the parallel channel was set to 1 mm, because the length and position of the tissue had variations at a sub-millimeter level that were caused by manual operations (i.e., cut and placement). The height of the parallel channel was 1.5 mm to prevent the tissue and any bubbles in the water from closing the parallel channels (Figure 3b). Polydimethylsiloxane (PDMS) was selected as the material for the H-shaped channel device because it should be biocompatible, flexible, and transparent.

The holders had holes around the H-shaped channel device to allow for the connection to syringe pumps and were screwed together to create a mechanical seal. In order to observe the tissue during the experiments, a transparent acrylic plate was used as the holder material.

### 2.3. Fabrication Process

The H-shaped channel device was fabricated using conventional soft lithography. A mixture of PDMS (Silpot184 W/C, Dow Corning Toray, Tokyo, Japan; main agent and hardener at a mass ratio of 10:1) was deaerated in a vacuum container and poured into molds. The PDMSs were cured at 100 °C for 60 min. The PDMS replica was demolded. The molds were fabricated from a polyacetal plate using a numerical control cutting machine (MDX-40A, Roland DG, Shizuoka, Japan). An end mill that was 0.5 mm in diameter was used. After cutting, the surface of the mold was polished down to 0.3 μm in roughness using sandpaper and an abrasive compound. The acrylic holder was fabricated using a laser machine (L906PC, IIDA, Aichi, Komaki, Japan) to create the holes. A tissue was manually cut from a frozen pig’s heart using a biopsy punch and the size was then adjusted.

### 2.4. Assembly Process

The tissue was set in the connection channel of the lower layer of the H-shaped channel device. The upper layer of the H-shaped channel device was aligned and placed on top of the lower layer. The aligned H-shaped channel device was pressed and screwed in place between the acrylic holders. Four screws were used to minimize the time required for the assembly in order to prevent the tissue from drying.

#### 2.4.1. Experimental Setup

The assembled device was set on the stage of an inverted microscope (CKX41, Olympus). The inlets of the parallel channels were connected to the syringe pumps with silicone tubes. DI water and blue-dyed water, degassed in a vacuum container for 24 h, were introduced into the H-shaped device. The flow rate was 1 μL/min for perfusion. Images of the tissue inside the device were taken every 1 h.

Here, we used food dye (Food coloring Blue, Kyoritsu Foods, Saitama, Soka, Japan) for the blue-dyed water. The dye contained Brilliant Blue FCF with 792.86 in molecular weight as the blue pigment. The diffusion coefficient, *D*, of Brilliant Blue FCF was calculated using the following equation:(1)$$D=\frac{kT}{6\pi \mu a}$$
where k is the Bolzmann constant, T is the absolute temperature, μ is the viscosity of water, and a is the radius of the spherical particle. T was 300 K in our experiment. μ was 0.85 × 10^−3^ Pa·s. The radius of a molecule of Brilliant Blue FCF was calculated to be 0.6 nm using the relation: weight of a single molecule = molecular weight/Abogadro constant = density × spherical volume of a molecule (4πa^3^/3). With the radius, the diffusion coefficient of the Brilliant Blue FCF against water was calculated to be 4.3 × 10^−10^ m^2^/s.

#### 2.4.2. Sectioning

To analyze the content of the tissue, the tissue was sectioned. The tissues were fixed with 4% paraformaldehyde for 24 h in the microfluidic device. A chamber in a cryostat (HM550, Thermo Fisher, Waltham, MA, USA) was cooled down to −20 °C. After the perfusion experiment, the microfluidic device was transported to that chamber of the cryostat and the top layer of the microfluidic device was opened. The tissue on the lower layer was then embedded with a cryo-compound (M.E CRYO COMPOUND, Microedge Instruments Inc. Surrey, BC, Canada), a type of embedding agent. The cryo-compound was solidified at −20 °C. The cryo-compound was pasted on a sample stage. On the solidified cryo-compound, the embedded tissue was transported and placed vertically, then embedded again in an extra cryo-compound. The tissue was sectioned in the radial direction, and the thickness of the slices was 50 μm.

#### 2.4.3. Image Analysis

The color of the tissue changed as a result of the prolonged perfusion of DI and blue-dyed water. To measure the blue intensity, the original image was divided into red, green, and blue components. The intensities of the red component, substracted from the original image, were measured. This was because the blue-dyed water included green and blue components. The blue intensity of the tissue was measured using the images of the tissue taken with a transmission microscope during and after perfusion. During perfusion, the blue intensity of the tissue was measured along a line in the longitudinal axis. In contrast, the blue intensity after perfusion was measured using images of the slices taken from the tissue. The blue intensities of the pixels inside the slice shape were measured and averaged. For the analyses of the image, we used the Image J software [31].

## 3. Results and Discussion

### 3.1. Fabricated Components and Assembly

The fabricated upper and lower layers of the H-shaped channel device are shown in Figure 4a,b, respectively. Parallel channels and connection channels were formed in both layers as a design. The fabricated holder was transparent and had holes for connection to the syringe pumps and screws (Figure 5). The tissue was a cylinder of 0.48 mm diameter and 3.8 mm length (Figure 6). The assembled device is shown in Figure 7. The tissue in the assembled device was observed through the device and the acrylic holder (Figure 7a). With four screws, we obtained sufficient contact to avoid liquid leakage between the upper and lower layers of the H-shaped channel device (Figure 7b).

### 3.2. Concentration Gradient Formed by Long-Term Perfusion and Real-Time Observation

To confirm that the H-shaped channel device could feed the culture medium to a piece of tissue for a long time, prolonged perfusion tests using water were performed (Figure 8). The upper and lower images are bright-field and red-substracted images, respectively. The tissue was in the connection channel, and DI and blue-dyed water were perfused in the parallel channels. One end of the tissue was exposed to the DI water and the other was exposed to the blue-dyed water. In the initial condition, the tissue was located in the connection channel, and then DI and blue-dyed water were introduced (Figure 8a). After exposure to the blue-dyed water, the area with high blue intensity (blue area) gradually increased (Figure 8b–f); this is because the blue dye in the water diffused into the tissue. After the perfusion experiment, the amounts of water collected from each outlet were 6.0 g. The DI and blue-dyed water were not mixed, leaked, or overflooded in the H-shaped channel device, especially through the connection channel, during the experiments (100 h). We also examined this performance using phosphate-buffered saline and Dulbecco’s modified Eagle medium (Appendix A).

The length of the blue area border from the junction between the connection and parallel channels (point A) was plotted as a function of time (Figure 9). At the initial stage of the experiments, the length of the blue area border increased dramatically. This means that the diffusion was fast. In the first twenty hours (0–20 h), the increase in the length of the blue area border was 0.36 mm. However, the increment of the length of the blue area border decelerated as time passed. In the last twenty hours (80–100 h), the increase in the length of the blue area border was 0.09 mm. This reduction in the rate of the diffusion is reasonable because the diffusion is in accordance with Fick’s law of time and the square of the distance. The approximate curve was L = 2.61t^1/2^ using the least-square method.

The diffusion coefficient can be calculated by L^2^/t, where L is the diffusion length and t is time. From this equation, the diffusion coefficient of the Brilliant Blue FCF in the tissue was 1.81 × 10^−12^ m^2^/s. The diffusion coefficient in the tissue was 2.4 × 10^2^ times smaller than the estimated coefficient in water. This could be because the density of the tissue was much higher than that of water. Furthermore, the diffusion coefficient in brain tissue was reported to be 1.3 × 10^−12^ m^2^/s [32], which was comparable to our results.

Bubbles appeared in the parallel channels during the experiment (Figure 8c–f). The bubbles were located at the top of the parallel channels, whose height was larger than that of the connection channel in order to keep the continuous flow in the parallel channels. Furthermore, the diffusion in the tissue follows Fick’s law. This means that the diffusion in the tissue did not stop during the experiment. We believe that the bubbles did not touch the ends of the tissue and therefore did not affect the contents or properties of the tissue.

### 3.3. Color Gradient Evaluation

#### 3.3.1. Intensity Distribution of Tissue from the Appearance

The tissue after prolonged perfusion is shown in Figure 10. At the beginning, the tissue was red (Figure 6). The red color of the tissue was bleached, possibly because the myoglobin in the tissue had disappeared. After prolonged perfusion, the end of the tissue that had been exposed to the blue-dyed water was colored blue (Figure 10a). The distribution of blue intensity in the longitudinal direction of the tissue after 100 h was measured (Figure 10b). Point B was set as the location of the junction between the connection channel and the parallel channels for the DI water (0.0 mm), and point C was the junction between the connection channel and the parallel channels for the blue-dyed water (3.0 mm). The median of the box plot shows that the blue intensity remained constant at almost zero from 0.0 to 1.8 mm and increased from 1.9 to 2.5 mm. The maximum blue intensity was 58.8 at 2.5 mm. From 2.5 to 3.0 mm, the intensity decreased. This is because the blue dye was highly concentrated at the end of the tissue, to the point that the hue turned black in the images, resulting in a decrease in the blue intensity. The large error bars may have been caused by the individualities of the tissue.

#### 3.3.2. Intensity Distribution of Tissue from Slices

After prolonged perfusion, the tissue was removed and sectioned. A series of the sectioned slices is shown in Figure 11. The thickness of the slices was 50 μm, and the number of the slices was 86. The average blue intensities inside the tissue along the longitudinal axis are shown in Figure 12. The positions of 0.0 and 3.0 mm corresponded to the points B and C, respectively. We focused on the median of the box plot, where the intensity obtained from the slice at 0.0 mm was 1.88. The intensity was almost constant at 2.0 mm. The intensity increased from 3.89 at 2.1 mm to 47.4 at 2.8 mm. Although the end that was exposed to the blue-dyed water was 3.0 mm, the intensity decreased to 25.5 from 2.9 to 3.0 mm due to the same reason as the intensity distribution of the tissue from the appearance. The large error bars may have been caused by differences in the tissue.

The positions of the peaks were different by 0.3 mm between the distributions of the appearance and content. Although the intensities were obtained from transmission microscopic images, the thicknesses of the samples differed between the appearance and content measurements (that is, the sample for the appearance measurement was a tissue of 0.5 mm thickness, and the sample for the content measurement was a slice of 50 μm thickness). Owing to the difference in sample thickness and the non-uniformity of the diffusion front in the tissue, the difference in the peak positions could be negligible. Therefore, the distributions are comparable.

## 4. Conclusions

We developed an H-shaped channel device to establish a concentration gradient in a tissue and analyze the gradient in the tissue from its appearance and content. We achieved the formation of a color gradient in the tissue by prolonged perfusion and tissue analysis. During the prolonged perfusion, the blue-dyed water gradually diffused into the tissue. The tissue was also sectioned to measure the intensity of the blue color in the tissue slices. We confirmed that a blue gradient was observed from the appearance and content of the tissue. In the future, we will conduct perfusion experiments and cultures of living cancer tissue, analyze the material diffusion and consumption using the Michaelis-Menten equation, and study the interaction between hypoxic and normoxic cells using a microfluidic device.

## Figures and Tables

**Figure 1 micromachines-12-01482-f001:**
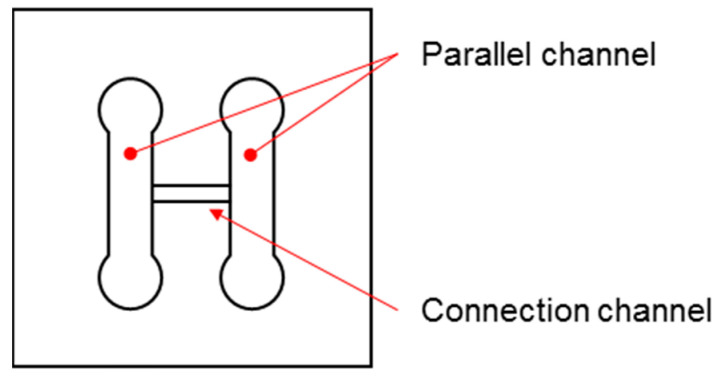
A schematic illustration of the H-shaped channel device. The H-shaped channel device has parallel channels to perfuse liquid and a connection channel to set a tissue under gradient.

**Figure 2 micromachines-12-01482-f002:**
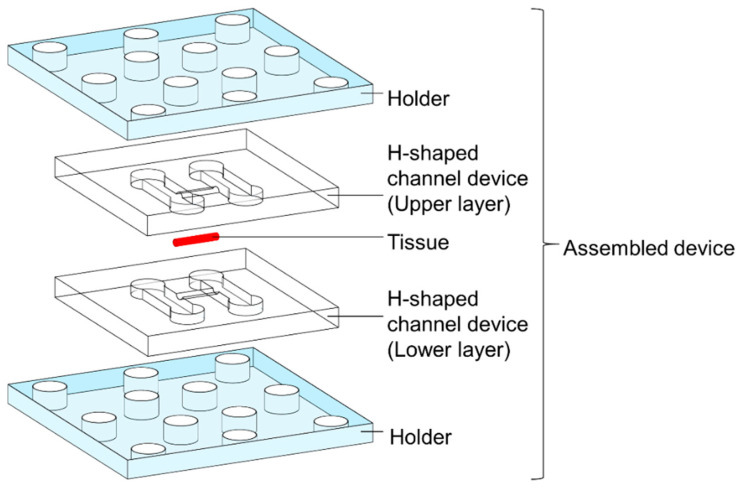
A schematic illustration of assembly. The tissue is set between the upper and lower layers of the H-shaped channel device. The H-shaped channel device is mechanically pressed by the holders to create a tight seal.

**Figure 3 micromachines-12-01482-f003:**
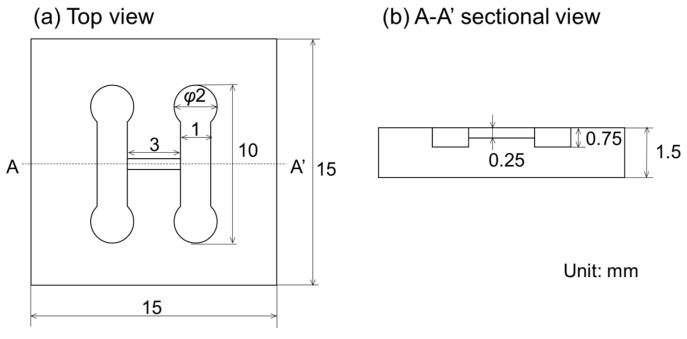
Design of the H-shaped channel device. (**a**) Top view. (**b**) A-A’ sectional view.

**Figure 4 micromachines-12-01482-f004:**
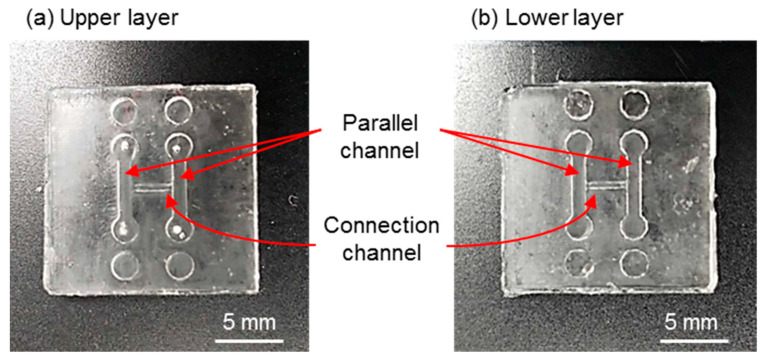
H-shaped channel device. (**a**) Upper layer. (**b**) Lower layer.

**Figure 5 micromachines-12-01482-f005:**
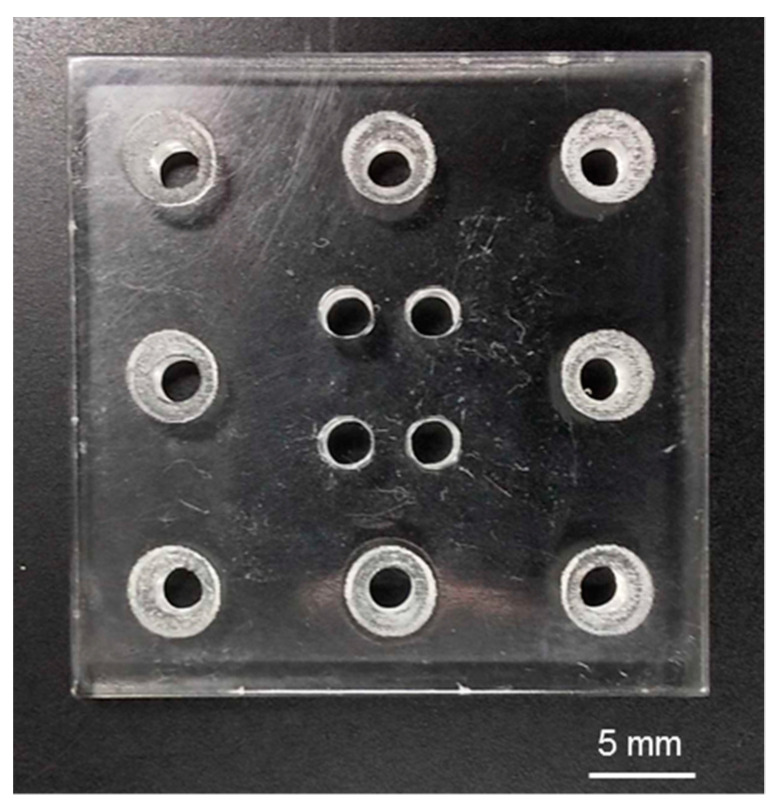
Acrylic holder. Four holes for connection to syringe pumps in the middle area and eight holes for screws around the edge.

**Figure 6 micromachines-12-01482-f006:**
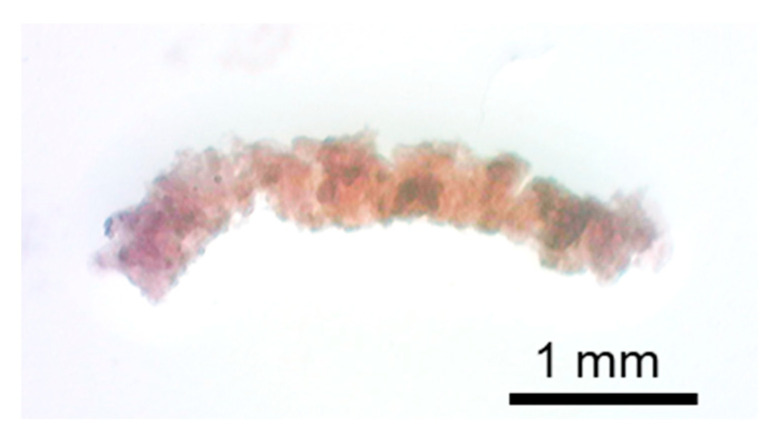
Tissue punched from the meat of a pig’s heart.

**Figure 7 micromachines-12-01482-f007:**
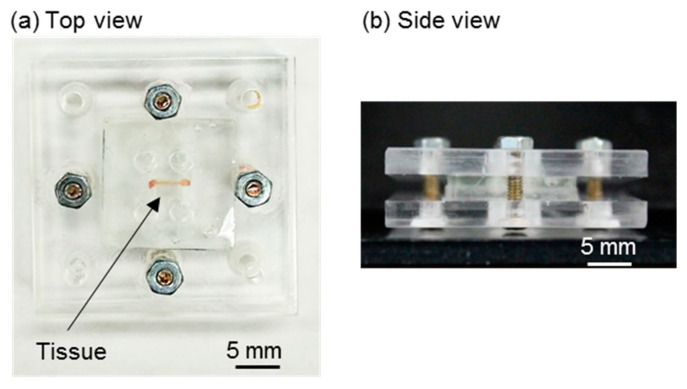
Assembled device. (**a**) Top view of the assembled device. The tissue can be observed owing to the transparency of the H-shaped channel device and the acrylic holder. (**b**) Side view of the assembled device.

**Figure 8 micromachines-12-01482-f008:**
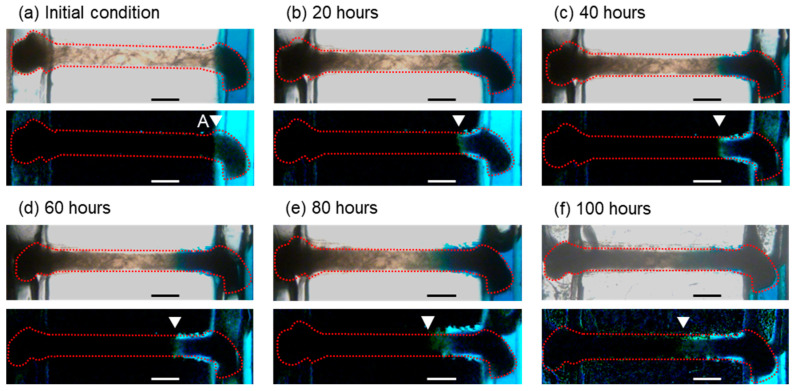
Tissue during prolonged perfusion. Upper and lower images are bright-field and red-substracted images, respectively. (**a**) Initial condition, (**b**) 20 h, (**c**) 40 h, (**d**) 60 h, (**e**) 80 h, and (**f**) 100 h later. Scale bar: 0.5 mm.

**Figure 9 micromachines-12-01482-f009:**
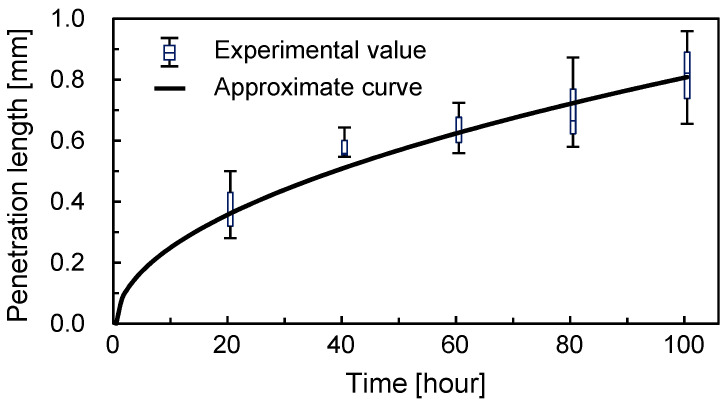
Temporal change in the diffusion length of the blue area in the tissue (N = 3). The approximate curve was L = 2.61t^1/2^.

**Figure 10 micromachines-12-01482-f010:**
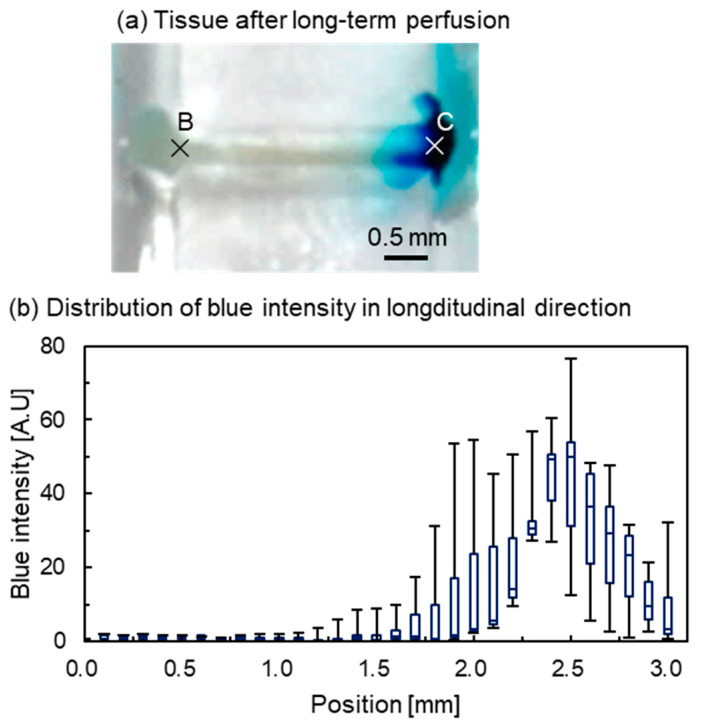
Distribution of blue intensity along the tissue after prolonged perfusion (N = 3). (**a**) Tissue after prolonged perfusion. It was bleached and colored blue on one side. (**b**) Distribution of blue intensity in the longitudinal direction. Points B and C correspond to 0.0 and 3.0 mm, respectively.

**Figure 11 micromachines-12-01482-f011:**
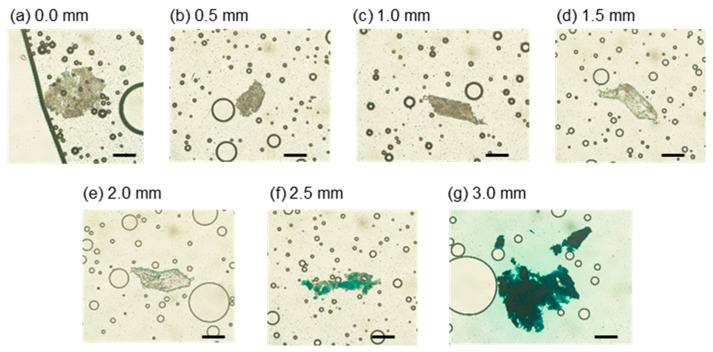
Series of slice images sectioned from tissue after long-term perfusion: (**a**) 0.0 mm, (**b**) 0.5 mm, (**c**) 1.0 mm, (**d**) 1.5 mm, (**e**) 2.0 mm, (**f**) 2.5 mm, (**g**) 3.0 mm. Scale bar: 0.2 mm.

**Figure 12 micromachines-12-01482-f012:**
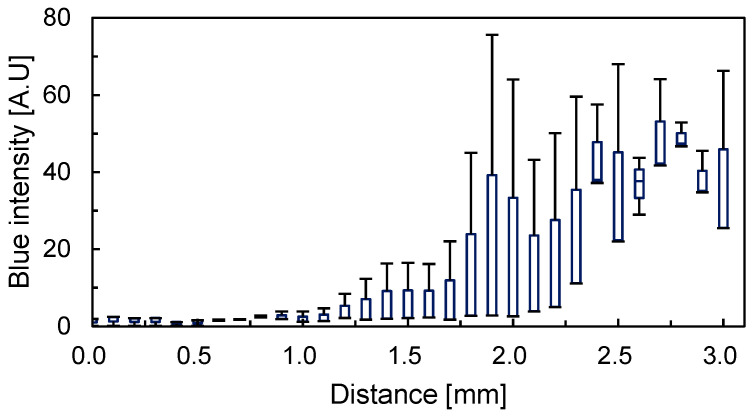
Blue intensities of the slices taken from the tissue as a function of distance (N = 3).

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
