# Peer review of "Development of a Microfluidic Device to Form a Long Chemical Gradient in a Tissue from Both Ends with an Analysis of Its Appearance and Content"

_micromachines, 2021, doi:10.3390/mi12121482_

Round 1
Reviewer 1 Report
The authors have assemble a very straightforward demonstration of a custom microfluidic device and performed a number of validation experiments on the main characteristic of interest, which is gradient formation. I appreciate the image quality of the tissue in the device and the cross-section validation of the same. In future experiments, cross-sections will be free important readout. While this does provide useful features (being able to take out tissue after experiment), the overall microfluidic device does not offer anything novel. However, given it's specific utility for a given experiment, it is deserving of publication in this journal after addressing the following comments:
A few English related changes:
- Abstract: "For demonstration, a cylindrical pork tissue specimen..." should be changed to read: "For demonstration, a cylindrical porcine tissue specimen..."
- Methods 2.2 (and fig.6 caption): "In this study, we used meat from a pig’s heart..." should be changed to read: "In this study, we used porcine cardiac muscle tissue..."
- all uses of the word "pork" should be changed to "porcine" as the proper scientific pig adjective
Overall comments:
- it would be nice if the authors could comment on if this device can be used with cell culture medium in sterile conditions so that live in vitro experiments can be maintained or if that is outside the scope.
- in this paper, the experimental design is based on a molecule that is not consumed by the tissue itself, therefore it's diffusion is purely based on max concentration driving the gradient. However in biological experiments with biological moleulcues diffusing, like with future hypoxia experiments where O2 is used by cells, the diffusion is based on the complex gradient of O2 usage along the hypoxic gradient, in which the Michaelis-Menten equation may need be applied. Also, with the biology of vasculature, capillaries are often within 100-200 microns of cells due to high O2 consumption. Given these considerations, micro tissues often reach hypoxic levels after 100-200 microns from the edge, therefore this device maybe physically too large for live tissue experiments using biological components (e.g., O2, drugs, proteins). Please comment on these limitations/considerations in the Discussion or Conclusion sections and how could this be addressed for future experiments.
Reviewer 2 Report
This manuscript propose an H-shaped channel device with a mechanical seal to perfuse the medium for 100 h to facilitate the formation of a concentration gradient in a tissue, and to analyze the tissue inside. The microfluidic device proposed in the manuscript has the advantages of simple operation and high repeatability. However, the innovation of the research content is not strong, and the research content is insufficient, and there are the following problems:
- Although the manuscript indicates that the device obtained sufficient contact to avoid liquid leakage between the upper and lower layers of the H-shaped channel device, there is a lack of testing and characterization of the device's leakage threshold.
- Both the research content and results of 3.2 and 3.3.1 are the tissues after prolonged perfusion,so the research contents and results are duplicated.
- Why are the error bars in the Figure 9, 10 and 12 so large? Does it indicate that the robustness of the device is poor?
- 6. The result of the tissue taken from the meat of pig heart is too vague, so it is recommended to replace it.
Reviewer 3 Report
The authors present a relatively straightforward method for experimental tissue perfusion, which is of increasing importance as tissue-scale models are increasingly used. However, it is not clear what the novel contribution to the field is. Gradient channels and flow channels have been demonstrated previously, and these specific experiments are minimal. The impact of this work could be improved through the use of living tissue and physiological media to demonstrate this can support perfusion for viable tissue. Further, from the experimental setup it is unclear if the tissue is being actively perfused. It seems more likely that convection only occurs is the main channels, not in the cross channel, reducing the usefulness of this culture device for maintaining metabolically active tissue in the cross channel. Experimental verification of both the flow through the tissue (for example, using fluorescent microspheres) and support of viable tissue (for example, using live/dead staining) would be needed . Finally, the authors make a point of the importance of removing tissue intact from the device, and indeed this can be a challenge in some microfluidic devices. To support this point, the authors should perform experiments to demonstrate that tissue morphology and structure is unaltered. The sections of tissue shown look deformed and of variable size, indicating possible damage during removal.
Round 2
Reviewer 2 Report
All the issues are addressed.
Reviewer 3 Report
The authors have presented a revised manuscript the clarifies and re-emphasizes the centimeter scale of the experiments. This is important, as noted in the original review. However, this basic technique (setting up a diffusion gradient with 2 or multichannel flow) has been demonstrated multiple times. For example, from a brief google search:
https://doi.org/10.1007/s10544-007-9051-9
https://doi.org/10.1039/B710524J
https://doi.org/10.1021/la7026835
And more as well. While the centimeter scale is important, and I haven't seen any devices of this kind at the centimeter scale, the concept is very straightforward scaling of prior work.
While I appreciate the authors additions, it is still necessary to show some experiments demonstrating that this device will be functional for the target application (e.g. tumor models or live tissue).